# SpikeZIP: Compressing Spiking Neural Network with Paths-Ensemble Training for Optimized Pareto-front Performance

## Abstract

Spiking neural network (SNN) has attracted great attention due to its great energy efficiency on neuromorphic hardware. By transferring the parameters of pretrained artificial neural network (ANN) and utilizing the ANN quantization, recent works of ANN-SNN conversion can produce SNNs with close-to-ANN accuracy and low inference latency (known as the number of time-steps). Nevertheless, existing works fail at providing theoretic equivalence between Quantized-ANN (QANN) and its converted SNN, while the SNN accuracy at small time-step (i.e. Pareto-frontier) can be further improved. To solve the problems, this paper proposes a novel conversion framework called SpikeZIP. The SpikeZIP utilizes the ANN-Quantized ANN(QANN)-SNN two-step conversion to obtain SNN which improves the Pareto frontier of accuracy versus inference time-steps. SpikeZIP integrates two novel algorithms: 1) a paths-ensemble training algorithm that considers the SNN temporal information when fine-tuning QANN; 2) a mathematically equivalent conversion algorithm between the whole QANN and SNN. In the experiment, SpikeZIP can achieve 73.92% accuracy on ImageNet with VGG-16 within 9 time-steps and 74.21% accuracy on ImageNet with ResNet-34 within 11 time-steps which are better than SOTA works. Spiking Neural Networks, Quantization, ANN-SNN Conversion

## 1 Introduction

Spiking neural network (SNN) Maass (1997) is a type of biologically plausible neural network inspired by brains of living organisms. Unlike modern artificial neural networks (ANNs) LeCun et al. (2015) use continuous activation value to propagate information between neurons synchronously, SNNs utilize discrete events or "spikes" for asynchronous neuron-to-neuron communication and processing Merolla et al. (2014); Davies et al. (2018). Such event-driven characteristic in SNN is considered as one of the key factors to achieve remarkable energy efficiency in human brain ($\sim$20W) Roy et al. (2019). Given the astonishing growing pace of computing power demands of ANN models (doubled per 2 months since 2020 Mehonic & Kenyon (2022)), evolving ANNs to energy-efficient SNNs is in urgent demand for cost-effective inference.

Currently, methods to train SNN come in twofold: *learning-* and *conversion-based* Roy et al. (2019). Both methods attempt to obtain SNNs with high accuracy and less time-step. Previous learning-based works leverage variants of spike-timing-dependent plasticity Diehl & Cook (2015) or gradient descent algorithms Wu et al. (2018; 2019); Neftci et al. (2019); Kim & Panda (2021); Zenke & Vogels (2021) to update the synaptic weights of SNN. Unfortunately, although learning methods train SNN at small time-step (e.g., 4 time-step), due to the inaccurate gradient approximation Neftci et al. (2019) for the non-differential SNN neuron, e.g., integrate and fire (IF) neuron, an accuracy gap persists between SNN and its ANN counterpart Fang et al. (2021a).

Rather than directly training an SNN, the conversion-based methods transfer the parameters of the pre-trained ANN into its SNN counterpart that yields close-to-ANN accuracy. As the accuracy versus time-step curve shown in fig. 1, the SOTA works (Offset Hao et al. (2023), QCFS Bu et al. (2023), etc.) reduce the inference time-step to 16. These works replace the ReLU function with its quantized version (*Q-ReLU*), and use a quantization-aware training algorithm to compress the quantization level of activation to reduce the inference time-step. However, the low quantization

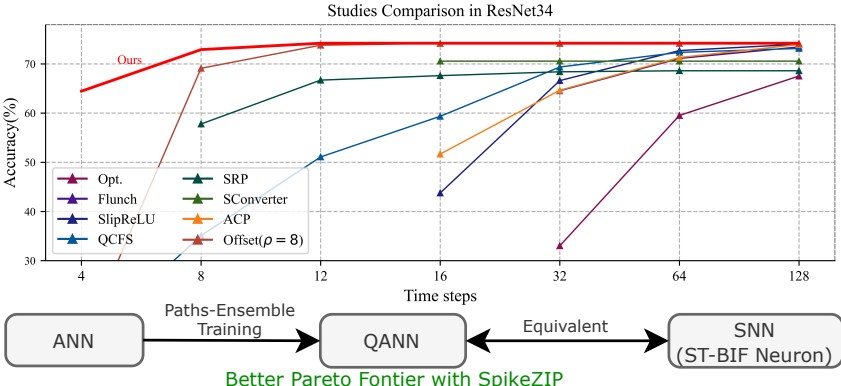

Figure 1: **SpikeZIP Highlights.** *Upper:* accuracy v.s. latency (#time-steps) curve of SNN (ResNet-34 on ImageNet) obtained from varying works. *Bottom:* conversion pipeline of SpikeZIP, which trains a QANN with paths-ensemble training and then converts the trained QANN to its mathematically equivalent SNN.

level of activation brings large quantization errors in ANN training, which sacrifices the accuracy of the ANN and the converted SNN. Therefore, improving the accuracy of SNN at small time-step by reducing the quantization level encounters bottleneck. For the neurons, such Q-ReLU approximates the conventional IF neuron and is equivalent to the sign neuron proposed in Hu et al. (2023); Wang et al. (2022a); Li et al. (2022). However, due to the max-pooling and batch normalization operator, non-equivalence still exists between Quantized-ANN and SNN at the model-level.

By far, two challenges remain unsolved: *1) non-existing work provides theoretic support of equivalence of QANN and its converted SNN at model level; 2) SNN accuracy at small time-step (e.g. Pareto frontier) can be further improved.* As the countermeasure, we build a framework called SpikeZIP to produce SNN with state-of-the-art performance (i.e., accuracy and latency). SpikeZIP fine-tunes QANN utilizing a pre-trained ANN, then the QANN is converted into its mathematically equivalent SNN. Our core technical contributions in SpikeZIP can be summarized as:

- **Theoretical Equivalence of QANN-SNN** is proved rigorously at the model level. Such equivalence holds under the condition of 1) ANN uses quantized ReLU; 2) SNN uses ST-BIF neuron model; 3) SNN-unfriendly operators are replaced (e.g., alter max pooling to average pooling); 4) taking analogRueckauer et al. (2017) input and bias encoding.

- **Pareto-front performance** is achieved in SpikeZIP-produced SNN. By incorporating a novel *Paths-Ensemble Training (PET)* technique during QANN training, the SNN converted from the QANN can obtain higher accuracy with lower latency (e.g. less number of time-steps), which is shown in fig. 1. In the experiment, the Spike-ResNet34 in SpikeZIP achieves 74.21% accuracy at 11 time-steps which is better than other state-of-the-art works (74.14% at 16 time-steps by Offset Hao et al. (2023)).

## 2 BACKGROUND AND RELATED WORK

**Neuron Model.** Recent works Li et al. (2022); Hu et al. (2023); Wang et al. (2022a) propose a variant of IF neuron, which we call bipolar integrate&fire with spike tracing (ST-BIF). Its neuron dynamics can be described as:

$$V_t = V_{t-1} + V_t^{\text{in}} - V_{\text{thr}} \cdot \Theta(V_{t-1} + V_t^{\text{in}}, V_{\text{thr}}, S_{t-1})$$

$$S_t = S_{t-1} + \Theta(V_{t-1} + V_t^{\text{in}}, V_{\text{thr}}, S_{t-1})$$

$$\Theta(V, V_{\text{thr}}, S) = \begin{cases} 1; & V \geq V_{\text{thr}} \ \& \ S < S_{\text{max}} \\ 0; & \text{other} \\ -1; & V < 0 \ \& \ S > 0 \end{cases} \quad (1)$$

where the notations used above are listed in table 1. ST-BIF neuron in eq. (1) differs from the vanilla IF neuron from two perspectives: 1) ST-BIF neuron fires bipolar spikes (either positive or negative);

Table 1: Summary of mathematical notations used in this paper.

| Notation | Description |
|---|---|
| $V_t$ | potential of neuron membrane at time-step $t$ |
| $V_{thr}$ | threshold voltage for neuron to fire a spike |
| $V^{in}, V^{out}$ | input or output voltage of neuron |
| $T_{eq}$ | time-steps of neuron enters equilibrium state |
| $T_{off}$ | time-steps when input and bias are turned off |
| $S_t, S_{max}$ | Spike tracer at time-step $t$ and maximum value of spike tracer |
| clip$(x, \alpha_{min}, \alpha_{max})$ | Clip function that limits $x$ between $\alpha_{min}$ and $\alpha_{max}$ |
| $\Theta(V, V_{thr}, S)$ | Output spike decision function of ST-BIF neuron |
| $\mathcal{F}_Q(\mathbf{X}_0|n, \{\mathbf{W}\}, \{\mathbf{s}\})$ | QANN with weights $\mathbf{W}$, quantization level $n$, scale $\mathbf{s}$ |
| $\mathcal{F}_S(\mathbf{V}|n, \{\mathbf{W}\}, \{\mathbf{s}\})$ | SNN with $S_{max} = n$, weight $\mathbf{W}$ and $V_{thr} = \mathbf{s}$ |
| $n, n_{mp}, n_{sp,i}$ | Quantization levels in major-path and $i$-th sub-path. |

Table 2: **Techniques and setting of related works.** ana. denotes analog input encoding, mem calib. is short for membrane calibration, and eq. is equivalence.

| | OPI | QCFS | EMRS | Radix | SNM | QFFS | Offset | Fast-SNN | Ours |
|---|---|---|---|---|---|---|---|---|---|
| (i) encoding | ana. | ana. | time | radix | ana. | ana. | ana. | ana. | ana. |
| (ii) neuron | IF | IF | OneSpike | LIF | ST-BIF | ST-BIF | IF | ST-BIF | ST-BIF |
| (iii) mem calib. | ✓ | | ✓ | | | ✓ | ✓ | ✓ | ✓ |
| (iv) Q-ReLU | | ✓ | | ✓ | | ✓ | ✓ | ✓ | ✓ |
| (v) neuron eq. | | | ✓ | ✓ | ✓ | ✓ | | ✓ | ✓ |
| (vi) model eq. | | | ✓ | ✓ | | | | | ✓ |

2) ST-BIF neuron guarantees the accumulated charge of output spikes equal to the output of the Q-ReLU function until the neuron stops firing, in virtue of the spike tracer $S_t$.

**ANN-SNN Conversion.** ANN-SNN conversion creates an SNN whose synaptic weights between neurons are identical to the corresponding ANN. Recent works focus on optimizing both the accuracy and latency of SNN (e.g. Pareto frontier in fig. 1), via leveraging the techniques (summarized in table 2) as follows:

*(i) Input encoding.* Four encoding methods are used most frequently, e.g., analog coding Rueckauer et al. (2017); Han et al. (2020); Li et al. (2022); Bu et al. (2023); Hao et al. (2023); Hu et al. (2023), rate coding Liu et al. (2022), time coding Park et al. (2020) and radix coding Wang et al. (2022b). *(ii) Neuron model.* The soft-reset IF neuron model Han et al. (2020) is the most common choice Han & Roy (2020); Han et al. (2020); Bu et al. (2022; 2023); Hao et al. (2023), which maintains the residual voltage after the neuron fire. To mitigate the occasional noise Li et al. (2022) between ANN and SNN, a sign neuron is proposed, which can fire negative spikes to offset the over-fired positive spikes Li et al. (2022); Wang et al. (2022a); Hu et al. (2023). Other neurons (e.g., iLIF Liu et al. (2022) and OneSpike Stanojevic et al. (2023)) are designed for specific systems, which are not common in recent works. *(iii) Membrane calibration* (e.g., initialize $V_0 = 0.5V_{thr}$ Bu et al. (2022)) has been widely adopted to reduce the ANN-SNN conversion error and latency. *(iv) Q-ReLU as ANN Activation* is a emerging trend whose converted SNN is found to have lower inference latency with decreased quantization levels Hao et al. (2023); Bu et al. (2023); Li et al. (2021; 2022); Deng & Gu (2021); Hu et al. (2023). *(v) Neuron equivalence* is proved Li et al. (2022); Wang et al. (2022a); Stanojevic et al. (2023); Hu et al. (2023) between SNN neurons (e.g., firing rate) and ANN activation functions (ReLU or Q-ReLU). *(vi) Model equivalence.* With time-based input, Stanojevic et al. (2023) claims the equivalent between ANN and SNN. For analog input, although Hu et al. (2023) claims the equivalence between QANN activation and SNN firing rate, there still exists accuracy degradation between QANN and SNN in experiments, which means the non-equivalence at the model level.

As summarized in table 2, although some works prove the equivalence at the neuron level Li et al. (2022); Wang et al. (2022a) and model level Stanojevic et al. (2023); Wang et al. (2022b), the equivalence between the QANN and SNN with analog encoding is missing.

**Quantization.** Quantization is a discretization process of the continuous value, which is widely used to compress ANN Gholami et al. (2022). The Q-ReLU function is written as:

$$f(x) = s \cdot \text{clip}(\text{round}(x/s), 0, n) \qquad (2)$$

where the notations used above are tabulated in table 1. To alleviate the accuracy degradation when quantizing with extremely low quantization level, LSQ Esser et al. (2020) is proposed to learn quan-

tization scale ($s$ in eq. (2)) during quantization-aware training Jacob et al. (2018); Gholami et al. (2022). LSQ has been used widely in previous conversion works Bu et al. (2023); Li et al. (2022); Hao et al. (2023); Hu et al. (2023), which is inherited by SpikeZIP as well.

# 3 METHODOLOGY

## 3.1 CONVERSION FLOW IN SPIKEZIP

SpikeZIP takes four steps in total to convert an ANN to its SNN counterpart. It firstly converts original ANN to SNN-friendly QANN through *(1) SNN-friendly morphing* and *(2) paths-ensemble training*. Then, SpikeZIP utilizes the *(3) operator fusion* and the *(4) neuron replacement* to QANN to obtain a high-performance SNN.

**1) SNN Friendly Morphing.** Since the original ANN may include operators and topology that are unfriendly to SNN, we modify those operators with the SNN-friendly counterparts. In detail, we alter the max-pooling to the average-pooling Rueckauer et al. (2017), as it is easier for equivalent SNN implementation. Moreover, we conduct residual connection re-routing (specified in section 3.3) for ResNet, to optimize the latency. Finally, the ReLU function is replaced by Q-ReLU function to convert ANN to QANN, for successive training.

**2) Paths-Ensemble Training** (PET) is proposed and utilized to train the morphed ANN (e.g. QANN) from the last step. PET (specified in section 3.4) is a variant of LSQ-based quantization aware training algorithm, which is designed to simultaneously improve the accuracy of trained QANN that quantized with varying quantization levels on the fly.

**3) Operator Fusion & 4) Neuron Replacing.** In operator fusion, SpikeZIP fuses the batch normalization layers of trained QANN into their front (e.g., convolution or linear) layers as in Chen et al. (2018). In the successive neuron replacing, we update all the Q-ReLUs to ST-BIF neurons, and adjust the configurations of ST-BIF neurons (e.g., $V_{t=0}$, $V_{\text{thr}}$, $S_{\text{max}}$ and etc., as discussed in proof 3.1) accordingly.

## 3.2 INPUT AND BIAS ENCODING

We design a unique analog encoding method for input and bias[1], which is also the prerequisite of QANN-SNN equivalence in section 3.5. Unliking traditional analog encoding that converts the raw image $\mathbf{X}_0$ to $\mathbf{V}_t^{\text{in}} = \mathbf{X}_0 \cdot \Delta v, \forall t$ and fires for each time-step, we choose the evenly-release strategy which sets $\mathbf{V}_t^{\text{in}} = \mathbf{X}_0 \cdot \Delta v / n, t \in \{1, 2, ..., T_{\text{off}}\}$. Hereby, input stops applying at $T_{\text{off}} = n$, where $n$ is the quantization level of QANN. Moreover, the evenly-release strategy is adopted for the bias in convolution/linear layers as well.

## 3.3 RESIDUAL CONNECTION RE-ROUTING

As the residual structure of ResNet He et al. (2016) plotted in fig. 3, the spiking neuron layer (SN, highlighted in orange) takes the addition of network blocks as its input, which makes it a "*spiking transmission bottleneck*" as more time-steps are taken for this neuron layer to integrate and fire.

As the countermeasure, we perform residual connection re-routing (RCR)[2] as described in fig. 3. If the residual connection is identity, we simply re-route the residual connection and addition after the Q-ReLU/SN (pink shadow part). If the residual connection uses convolution, in addition to the aforementioned rerouting process for identity, we insert an extra Q-ReLU/SN after the convolution.

---

[1]Biases are introduced due to the fused batch normalization, which can be viewed as the special form of inputs as they are constants as well.

[2]Prior SEW-ResNet Fang et al. (2021a) adopts "RELU before addition" (similar topology) to overcome vanishing or exploding gradient, during the direct training of SNN. In contrast, SpikeZIP is a conversion-based framework and RCR is used for latency reduction.

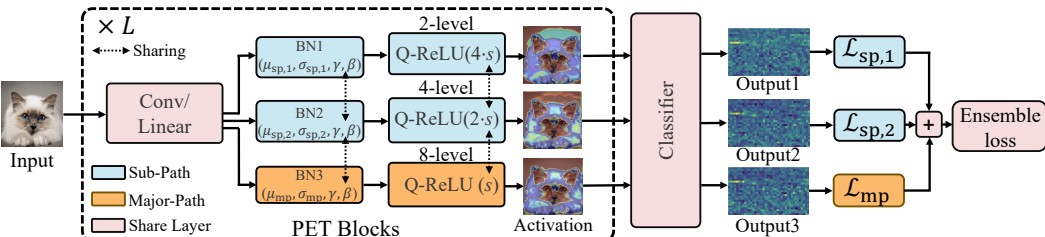

Figure 2: **Illustration of Paths-Ensemble Training (PET).** Given a QANN to be trained, PET introduces three paths (one major-path and two sub-paths). All three paths share identical parameterized layers (e.g., convolution/linear, etc.), but with independent Q-ReLU, batch-norm and loss. In one PET block, Q-ReLUs use shared quantization scale $s$, but scaled as $\{s, 2s, 4s\}$ for Q-ReLUs quantized with $\{8, 4, 2\}$-level respectively. For batch-norm Ioffe & Szegedy (2015), the mini-batch mean $\mu_i$ and variance $\sigma_i$ are collected for each path individually, while learned parameters $\gamma$ and $\beta$ are shared. Losses of each path are calculated independently and then added as one ensemble loss for training.

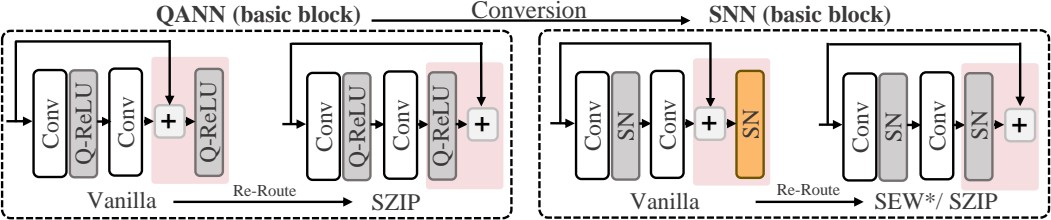

Figure 3: **Residual connection re-routing** (RCR) used to mitigate the spiking bottleneck phenomenon and reduce latency. SN denotes the spiking neuron (e.g., ST-BIF) layer. * SEW-ResNet Fang et al. (2021a) takes a similar topology but for different design purposes.

### 3.4 PATHS-ENSEMBLE TRAINING

PET is inspired by our observation from Bu et al. (2023); Li et al. (2022): *reducing the quantization levels of QANN lowers the latency of converted SNN by sacrificing accuracy as well.* As we expect the QANN-converted SNN can achieve higher accuracy at small time-steps (Pareto-front performance in fig. 1), the observation motivates us to improve the accuracy of QANN quantized with fewer quantization levels. Inspired by the slimmable network Yu et al. (2018) that trains CNNs with different channel numbers using shared parameters, we train QANNs quantized with varying quantization levels, using a *single set* of trained parameters (e.g., weights, bias and etc.).

**Training Strategy.** As illustrated in fig. 2, PET consists of three optimizing paths (one major-path and two sub-paths). Each path has its independent Q-ReLU and batch normalization (batch-norm) layer, while the remaining parametric layers (e.g. convolution, linear, etc.) are shared by the different paths. Note that, **only the major-path of QANN will be converted to SNN** once training with PET is completed, while sub-paths are used to assist the training in major-path. Three kinds of operators are carefully designed in PET:

*1) Parametric Layer:* The weights in parametric layers are shared among the paths in PET, which will be optimized by back-propagation to improve the accuracy of different paths simultaneously.

*2) Q-ReLU:* Assume the Q-ReLU in major-path is:

$$f_{\text{mp}}(x) = s \cdot \text{clip}(\text{round}(x/s), 0, n_{\text{mp}}) \tag{3}$$

Then, the Q-ReLU in $i$-th sub-path is written as:

$$f_{\text{sp},i}(x) = \frac{n_{\text{mp}}s}{n_{\text{sp},i}} \cdot \text{clip}(\text{round}(\frac{n_{\text{sp},i}x}{n_{\text{mp}}s}), 0, n_{\text{sp},i}) \tag{4}$$

where quantization levels are set as $\{n_{\text{mp}}, n_{\text{sp},1}, n_{\text{sp},2}\} = \{8, 4, 2\}$. The quantization scale $s$ is the learnable parameter that is shared between different Q-ReLUs.

*3) Batch-norm:* Since Q-ReLUs in different paths adopt varying quantization levels, the output distributions of Q-ReLUs are distinguished from each other significantly. According to our experiment, naively using one shared batch-norm makes it difficult for the training to converge. Therefore, we make the mini-batch mean $\mu_i$ and $\sigma_i$ be independent for each path, while learned parameters $\gamma$ and $\beta$ are shared. The modified batch-norm can be expressed as:

$$\text{BN}_{\text{mp/sp},i}(x) = \frac{x - \mu_{\text{mp/sp},i}}{\sigma_{\text{mp/sp},i}}\gamma + \beta \tag{5}$$

**Ensemble Loss.** The loss function can be described as:

$$\mathcal{L} = \mathcal{L}_{\text{mp}} + \alpha \sum_{i=1}^{2}(n_{\text{sp},i}/n_{\text{mp}}) \cdot \mathcal{L}_{\text{sp},i} \tag{6}$$

where $\mathcal{L}_{\text{mp}}$ and $\mathcal{L}_{\text{sp},i}$ are the cross-entropy loss of major-path and sub-paths respectively. $\alpha$ is the coefficient to scale the loss. To maximize the accuracy of QANN, the distillation method in LSQ Esser et al. (2020) is utilized, where the teacher network is the full precision ANN.

## 3.5 EQUIVALENCE OF QANN AND SNN

The QANN-SNN equivalence is the foundation behind the conversion algorithm from QANN to SNN in SpikeZIP. Notations are summarized in table 1. Assume the external stimulate (e.g., input and bias) are applied to SNN from $T = 0$ to $T_{\text{off}}$, we define the **equilibrium state** of SNN as the status where neurons of entire SNN are static (e.g., no further activities of neuron firing and membrane update). The time-step that SNN enters the equilibrium state is noted as $T_{\text{eq}}$.

**Lemma 1** *After entering the equilibrium state at $T_{\text{eq}}$, the accumulated output spikes of one ST-BIF neuron can be derived as a closed-form equation of quantization function:*

$$V^{\text{out}} = V_{\text{thr}} \cdot \text{clip}(\text{floor}(\frac{V^{\text{in}} + V_{t=0}}{V_{\text{thr}}}), 0, S_{\text{max}}) \tag{7}$$

*where $V^{in} = \sum_{t=0}^{T_{eq}} V_t^{in}$ is the accumulated input until $T_{eq}$, and $V_{t=0}$ denotes the initial membrane potential.*

lemma 1 shows the equivalence between ST-BIF neuron and Q-ReLU function. The detailed proof of lemma 1 can be found in previous work Hu et al. (2023).

**Theorem 1** *Assume a QANN $\mathcal{F}_{\text{Q}}$ parameterized by quantization level $n$, synaptic weights $\{\mathbf{W}_l\}$, quantization scale $\{\boldsymbol{s}_l\}$ and an SNN $\mathcal{F}_{\text{S}}$ converted from the QANN, the accumulated outputs of the SNN is equal to the QANN output:*

$$\mathcal{F}_{\text{Q}}(\mathbf{X}_0 \mid n, \{\mathbf{W}_l\}, \{\boldsymbol{s}_l\}) = \sum_{t=0}^{T_{\text{eq}}} \mathcal{F}_{\text{S}}(\{\mathbf{V}_t^{\text{in}}\} \mid n, \{\mathbf{W}_l\}, \{\boldsymbol{s}_l\}) \tag{8}$$

*where $\mathbf{X}_0$ represents the input for ANN, $\{\mathbf{V}_t^{\text{in}}\}$ denotes the encoded analog input (introduced in section 3.2) for SNN.*

**Proof 3.1** *Suppose $\{$convolution, batch-norm, Q-ReLU$\}$ is the $l$-th block of the network, convolution is parameterized by trained weight $\mathbf{W}_l$ and bias $\boldsymbol{b}_l$. With the batch-norm fused into convolution via operator fusion Chen et al. (2018), the weight and bias are updated as $\hat{\mathbf{W}}_l$ and $\hat{\boldsymbol{b}}_l$ respectively. Then, according to eq. (2), the output of such block $\mathbf{X}_l$ is written as:*

$$\mathbf{X}_l = s \cdot clip(round(\frac{\hat{\mathbf{W}}_l \cdot \mathbf{X}_{l-1} + \hat{\boldsymbol{b}}_l}{s}), 0, n) \tag{9}$$

*Considering the $l$-th block of SNN converted from $l$-th layer of QANN, by setting $\mathbf{V}^{in} = \hat{\mathbf{W}}_l \cdot \mathbf{X}_{l-1} + \hat{\boldsymbol{b}}_l, V_{t=0} = 0.5V_{thr}, S_{max} = n, V_{thr} = s$, eq. (7) and eq. (9) are equivalent. By extending the equivalence between blocks to the network, the eq. (8) is proven.*

Table 3: **Comparison with previous conversion-based methods.** Column of (Q)ANN *acc.* lists the validation accuracy achieved by its corresponding ANN or QANN. Column of *acc./T* lists the accuracy of SNN acquired in different works and the earliest time-step when reaches the accuracy. *n/a* refers to data not reported or cannot be reproduced; $^{\triangle}$ indicates the SNN enters the equilibrium state; – denotes most of the spikes have not reached the output of SNN. Current and prior best results are in **bold** and grey respectively. † Due to the Offset Hao et al. (2023) method requires $\rho$ time-steps to calculate offset spikes, the time-steps of Offset work spend should add $\rho$ ($\rho = 4$ on CIFAR100 and $\rho = 8$ on ImageNet). * denotes the results reproduced from the code.

| Method | (Q)ANN Acc. | Acc./T | SNN Accuracy | | | |
|---|---|---|---|---|---|---|
| | | | $T$=2 | $T$=4 | $T$=8 | $T$=16 |
| OPIBu et al. (2022) | 76.31 | 74.82/32 | n/a | n/a | 60.49 | 70.72 |
| QCFSBu et al. (2023) | 76.28 | 77.01/32 | 63.79 | 69.62 | 73.63 | 76.24 |
| SlipReLUJiang et al. (2023) | 70.03 | 70.65/32 | 58.66 | 62.56 | 66.31 | 69.35 |
| Fast-SNNHu et al. (2023)* | 68.16 | 66.74/16 | 58.09 | 65.53 | 66.67 | 66.74 |
| OffsetHao et al. (2023)† | 76.28 | 76.96/36 | n/a | 74.24/5 | 76.26/8 | 76.77/20 |
| SpikeZIP-P | 77.07 | **77.21/9** (0.25↑/27↓) | **64.63** (0.84↑) | **75.19** (0.95↑) | **77.02** (0.76↑) | 76.99 (0.22↑) |
| OPIBu et al. (2022) | 70.43 | 67.18/32 | n/a | n/a | 23.09 | 52.34 |
| QCFSBu et al. (2023) | 69.94 | 69.82/32 | 19.96 | 34.14 | 55.37 | 67.33 |
| SlipReLUJiang et al. (2023) | 68.40 | 68.76/32 | 23.79 | 37.94 | 57.20 | 66.61 |
| OffsetHao et al. (2023)† | 69.97 | 70.29/36 | n/a | 59.22/5 | 65.18/8 | 69.44/20 |
| Fast-SNNHu et al. (2023)* | 68.08 | 68.67/8 | 37.88 | 62.39 | 68.12 | 68.08 |
| SpikeZIP-PR | 70.10 | **70.31/10** (0.02↑/26↓) | **44.90** (7.02↑) | **63.58** (1.19↑) | **68.65** (0.53↑) | **70.07** (0.63↑) |

(a) **Comparison on CIFAR100**. *Upper* : results with VGG-16; *Bottom* : results with ResNet-20.

| Method | (Q)ANN Acc. | Acc./T | SNN Accuracy | | | |
|---|---|---|---|---|---|---|
| | | | $T$=2 | $T$=4 | $T$=8 | $T$=16 |
| OPIBu et al. (2022) | 74.85 | 74.69/512 | n/a | n/a | 6.25 | 36.02 |
| QCFSBu et al. (2023) | 74.29 | 74.32/1024 | n/a | n/a | 19.12 | 50.97 |
| SlipReLUJiang et al. (2023) | 71.99 | 72.02/128 | n/a | n/a | n/a | 51.54 |
| QFFSLi et al. (2022) | 73.08 | 73.10/8 | n/a | n/a | 73.10 | n/a |
| Fast-SNNHu et al. (2023)* | 73.02 | 73.29/16 | 52.69 | 71.13 | 72.94 | 73.29 |
| OffsetHao et al. (2023)† | 74.19 | 73.82/16 | n/a | n/a | 63.84/9 | 73.82/16 |
| SpikeZIP-P | 73.90 | **73.92/9** (0.12↑/7↓) | **59.04** (6.35↑) | **72.27** (1.14↑) | **73.75** (0.65↑) | **73.90** (0.08↑) |
| QCFSBu et al. (2023) | 74.32 | 73.37/256 | n/a | 12.75 | 35.06 | 59.35 |
| SlipReLUJiang et al. (2023) | 75.08 | 74.01/128 | n/a | n/a | n/a | 43.76 |
| OffsetHao et al. (2023)† | 74.22 | 74.14/16 | n/a | n/a | 69.11/9 | 74.14/16 |
| SpikeZIP-PR | 73.85 | **74.21/11** (0.07↑/5↓) | **15.39** (15.39↑) | **64.46** (51.71↑) | **72.91** (3.81↑) | 73.91 (0.25↓) |

(b) **Comparison on ImageNet**. *Upper* : results with VGG-16; *Bottom* : results with ResNet-34.

## 4 EXPERIMENT

We conduct experiments using the image classification task with VGG16 Simonyan & Zisserman (2014) and ResNet20 He et al. (2016) on CIFAR100 Krizhevsky et al. (2009), while VGG16 and ResNet34 on ImageNet Russakovsky et al. (2015). Experimental setups are specified in the appendix. The SpikeZIP has four variants of settings: without PET and RCR (SpikeZIP-N), with PET (SpikeZIP-P), with RCR (SpikeZIP-R), and with both PET and RCR (SpikeZIP-PR).

### 4.1 COMPARISON WITH PREVIOUS RESULTS

**Comparisons with conversion-based methods,** including OPI Bu et al. (2022), QCFS Bu et al. (2023), Offset Hao et al. (2023), QFFS Hu et al. (2023), SlipReLU Jiang et al. (2023), Fast-SNN Hu et al. (2023), are tabulated in table 3. We use SpikeZIP-P[3] and SpikeZIP-PR for VGG16 and ResNet20/34 respectively. We observe SNNs generated by our SpikeZIP achieve performance of Pareto frontiers, e.g., accuracy is higher than competing works across various time-steps. As listed in the column of *acc./T*[4] in table 3, *SpikeZIP significantly reduce the required time-step while not sacrificing the accuracy*. For example, compared to prior SOTA results, SpikeZIP reduce time-steps by 7 (43.75% reduction) for VGG16 and 5 (33.3% reduction) for ResNet34, both on ImageNet. Note that, at each time-step, since we use a single SNN generated by SpikeZIP to compete with all previous works (several works specially optimize the accuracy at lower time-step by compromising the upper-bound accuracy), some accuracy improvement of SpikeZIP does not seem significant.

**Comparison with learning-based methods** is elaborated in table 4 to show the advantages of the conversion-based method in SpikeZIP. In experiments, we set quantization level $\{n_{\mathrm{mp}}, n_{\mathrm{sp},1}, n_{\mathrm{sp},2}\} = \{3, 2, 1\}$ for PET in table 4. When inference time-step $T$ is 4, SpikeZIP outperforms the MS-ResNet Hu et al. (2021) and SEW Fang et al. (2021a) (prior typical learning-based works in CNN-based network) with 0.39% and 1.93% accuracy enhancement on ImageNet.

Moreover, fig. 4 depict that, as SEW Fang et al. (2021a) (trained by BPTT Zenke & Vogels (2021); Wu et al. (2019)) optimizes SNN to inference at $T$=4 specifically, its accuracy versus time-step curve approach the peak accuracy at $T$=4 then the accuracy goes downhill. In addition, we plot the training cost of SpikeZIP-PR, SpikeZIP-R and SEW with GPU hours as the evaluation metric. Although PET increases the training cost slightly, it takes much less ($\sim 43.5\times$ reduction) of computational resources to achieve even better accuracy. Such training cost reduction benefits from 1) parameters

---

[3]VGG has no residual connection, thus RCR is not applicable.

[4]The peak accuracy of SNN is not achieved at the $T_{\mathrm{eq}}$ but a time-step $T < T_{\mathrm{eq}}$, which has been observed in many previous works Li et al. (2022); Hao et al. (2023).

| Method | Type | Model | Param. | *Acc.* | T |
|---|---|---|---|---|---|
| HC-STDB Rathi et al. (2020) | hybrid | ResNet34 | 21.79 | 61.48 | 250 |
| DSR Meng et al. (2022) | supervised | PreAct-ResNet18 | 21.79 | 67.74 | 50 |
| PLIF Fang et al. (2021b) | BPTT | ResNet34 | 21.79 | 67.04 | 7 |
| STBP-tdBN Zheng et al. (2021) | BPTT | ResNet34 | 21.79 | 63.72 | 6 |
| TET Deng et al. (2022) | BPTT | ResNet34 | 21.79 | 64.79 | 6 |
| SEW Fang et al. (2021a) | BPTT | ResNet34 | 21.79 | 67.04 | 4 |
| MS-ResNetHu et al. (2021) | BPTT | ResNet34 | 21.80 | 69.40 | 4 |
| SpikeZIP-PR[†] | BPTT | ResNet34 | 21.79 | **69.79** | 4 |

Table 4: Comparison with learning-based methods.

Figure 4: Training Cost Comparison

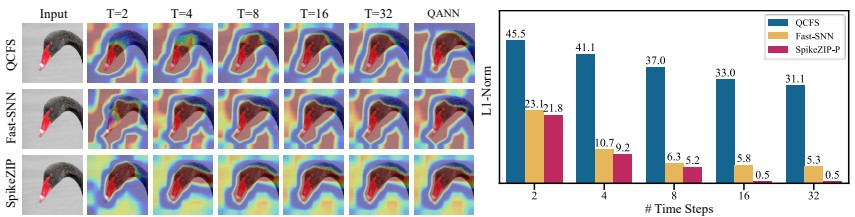

Figure 5: **Comparison of feature maps with conversion-based works,** using VGG16 on ImageNet. (Left) evolution of feature maps (e.g. accumulated spikes) w.r.t time-steps. (Right) L1-norm between accumulated feature map of SNN and feature map of QANN, using 100 image samples from ImageNet. SpikeZIP takes fewer time-steps and acquires an identical feature map of QANN.

of pretrained ANN are inherited for QANN fine-tuning; 2) BPTT introduces an extra time-dimension to the input and activation tensors which consumes more GPU memory and more number of GPUs.

## 4.2 Feature Map in QANN and SNN

To further demonstrate that SNN generated by SpikeZIP is functionally equivalent to QANN and takes fewer time-steps for SNN inference, we visualize the evolution of feature maps w.r.t different time-steps $T$, as depicted in fig. 5. The feature maps are obtained by accumulating the fired output spikes of SNN neurons which corresponds to the pixels in the visualized feature map. Two previous works compared here are QCFS Bu et al. (2023) and Fast-SNN Hu et al. (2023), where Fast-SNN claims the equivalence between the activations in QANN and SNN. From fig. 5, we can draw the following observations visually and quantitatively: 1) SpikeZIP takes less number of time-steps for accumulated feature maps of SNN to evolve close to QANN; 2) Since the neuron level inequivalence in QCFS Bu et al. (2023) (IF neuron is not equivalent with Q-ReLU) and model level inequivalence in Fast-SNN Hu et al. (2023) (lacking the concept of equilibrium state), the L1 distances of QCFS and Fast-SNN remain at a large value in fig. 5 which is supposed to be close to 0 for absolute model level equivalence. Compared to them, *only SpikeZIP shows the equivalence* both theoretically (section 3.5) and experimentally e.g., L1 distance is 0.5. Note that, calculating the L1-norm as 0.5 is resulted from the intrinsic computing error of GPU hardware.

## 4.3 Ablation Study

**PET and RCR.** To investigate the effectiveness of PET and RCR in SpikeZIP, we plot the curves of accuracy versus time-step when PET and RCR are adopted. As depicted in fig. 6a, both PET (-P) and RCR (-R) significantly enhance the inference accuracy of SNN at small time-step. Combining them together (e.g., SpikeZIP-PR) on ResNet, a further accuracy boost is observed, which demonstrates the compatibility between PET and RCR. Note that, there exists a small accuracy gap between the native ANN and peak accuracy achieved by the SpikeZIP variants. This is mainly resulted from the low quantization levels (e.g., $n_{mp} = 4$ used for ResNet20 on CIFAR10) chosen to train the QANN for a fair comparison with the competing works. Such an accuracy gap can be easily eliminated by relaxing $n_{mp}$ to a carefully selected but relatively large value.

**Quantization levels in PET.** Choosing different quantization levels for QANN during PET in SpikeZIP can lead to varying trends of accuracy versus time-step trade-off in its converted SNN. As

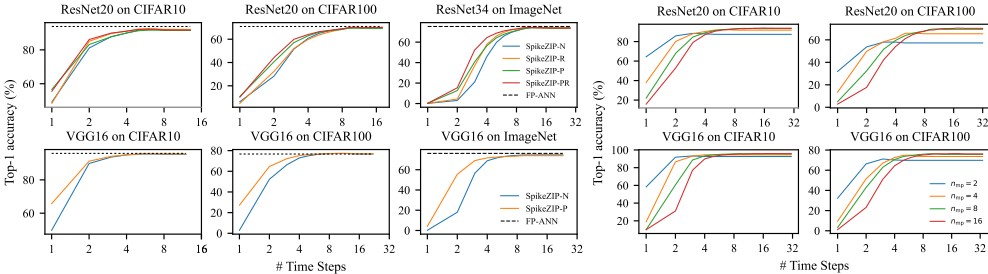

(a) Ablation of PET and RCR in SpikeZIP.   (b) Ablation of quantization levels $n_{mp}$.

Figure 6: **Curves of accuracy v.s. time-steps** for (a) SpikeZIP equips with varying techniques (e.g., N, R, P, PR); (b) PET using different quantization levels $n_{mp}$ for the main-path. The black dash-line labels the accuracy of native floating-point ANN.

Table 5: **Ablation Studies of SpikeZIP using ResNet34 on ImageNet.** (a) quantization scale and (b) batch-norm are ablated with three schemes of parameters sharing: {*independent*, *identical*, *share*}. Column of *independent* (*indep.*) scheme reports the accuracy of SNN major-path. (c) loss coefficient and (d) loss label type in the loss function. Grey indicates the default settings.

| $T$ | indep. | identical | share | $T$ | indep. | identical | share | $T$ | $\alpha=0.25$ | $\alpha=1$ | $\alpha=5$ | $\alpha=10$ | $T$ | Hard | Soft | Hard+Soft |
|---|---|---|---|---|---|---|---|---|---|---|---|---|---|---|---|---|
| 2 | 1.364 | 0.352 | 15.39 | 2 | 9.662 | 43.08 | 15.39 | 2 | 3.548 | 6.154 | 17.06 | 21.38 | 2 | 16.49 | 15.39 | 16.77 |
| 4 | 52.65 | 32.88 | 64.46 | 4 | 61.62 | 66.00 | 64.46 | 4 | 55.77 | 59.29 | 63.15 | 63.37 | 4 | 64.61 | 64.46 | 64.71 |
| 8 | 72.01 | 71.43 | 72.91 | 8 | 72.86 | 66.87 | 72.91 | 8 | 71.93 | 72.11 | 71.59 | 71.37 | 8 | 72.90 | 72.91 | 72.92 |
| 16 | 72.46 | 71.61 | 73.91 | 16 | 73.94 | 68.79 | 73.91 | 16 | 73.29 | 73.14 | 72.78 | 72.62 | 16 | 73.76 | 73.91 | 73.93 |
| QANN | 72.32 | 71.58 | 73.85 | QANN | 74.06 | 68.88 | 73.85 | QANN | 73.30 | 73.06 | 72.72 | 72.46 | QANN | 73.76 | 73.85 | 73.76 |
| (a) $s$ in Q-ReLUs | | | | (b) $\{\mu, \sigma\}$ in BN | | | | (c) $\alpha$ in loss function | | | | | (d) Label for $\mathcal{L}_{sp,i}$ | | | |

shown in fig. 6b, $n_{mp} \in \{2, 4, 8, 16\}$ are investigated. We can conclude that, using smaller $n_{mp}$ can improve the accuracy of SNN at low time-step, while sacrificing the accuracy at high time-step (e.g., $T > T_{eq}$). When a greater $n_{mp}$ is adopted, it takes more time-steps for SNN to achieve a competitive accuracy that is close to the floating-point ANN counterpart.

**Parameter Sharing in PET.** As we specified in section 3.4, PET tactfully performs parameter sharing among Q-ReLUs (quantization scales $s$) and batch-norm ($\mu, \sigma$}) layers belonging to different paths. We perform the ablation study with three schemes, 1) *independent*: all trainable parameters in Q-ReLUs and batch-norms are independent from each other; 2) *identical*: paths use the identical set of parameters; 3) *share*: parameters are shared but using the proper scaling in eqs. (3) and (4) and partial sharing in eq. (5).

Ablation of parameter sharing in Q-ReLU is reported in table 5a. We can find that, incorporating the shared $s$ with properly scaled for different paths (e.g., *share* scheme) achieves the best accuracy across varying time-steps $T$, while the *identical* and *independent* show the worse performance due to the invalidation of sharing rule in the conversion theory. Ablation of parameter sharing in batch-norm is tabulated in table 5b. In contrast to the *share* scheme that pioneers the accuracy at all time-steps for Q-ReLU in table 5a, applying the *share* scheme on batch-norm layers performs better than the *independent* scheme at low time-step while outperforms the *identical* scheme at high time-step.

**Ensemble loss in PET.** In eq. (6), we design a additive loss from all paths. We first investigate the effect of tuning loss coefficient $\alpha$ from 0.25 to 10, as listed in table 5c. Choosing a larger $\alpha$ plays a role of encouraging SNN (major-path) to achieve higher accuracy at low time-step. On the contrary, a smaller $\alpha$ makes the loss function weigh more on the loss term that optimizes the major-path and increases the accuracy of SNN after $T_{eq}$. We take $\alpha = 1$ as the default setting.

Furthermore, we also examine the benefit of using a hard label (e.g., ground-truth label to calculate cross-entropy), soft label (e.g., logits from teacher ANN to calculate KL divergence), or mixed fashion, for the loss term of sub-paths $\mathcal{L}_{sp,i}$. Experiments in table 5d shows purely leveraging the soft-label for $\mathcal{L}_{sp,i}$ leads to the best accuracy of QANN (main-path), which is the default setting.

### 4.4 EXPERIMENTS ON OBJECT DETECTION

To show the potential of the conversion theory of SpikeZIP, we employ the SpikeZIP to YOLOv3 Redmon & Farhadi (2018) on object detection tasks. To convert the YOLOv3 to SNN, we first

Table 6: **Performance comparison for object detection task on PASCAL VOC 2007 and MS COCO 2017 between SpikeZIP and other conversion-based works.** SpikeZIP achieves higher mAP and less ΔmAP than the Fast-SNN Hu et al. (2023).

| Work | PASCAL VOC 2017 | | | | | MS COCO 2017 | | | | |
|---|---|---|---|---|---|---|---|---|---|---|
| | Architecture | ANN mAP | SNN mAP | ΔmAP | Time Steps | Architecture | ANN mAP | SNN mAP | ΔmAP | Time Steps |
| Spike-YOLO | Tiny YOLO | 53.01 | 51.83 | -1.18 | 8000 | Tiny YOLO | 26.24 | 25.66 | -0.58 | 8000 |
| Spike-YOLO V2 | Tiny YOLO | 53.01 | 51.74 | -1.27 | 5000 | Tiny YOLO | 26.24 | 25.78 | -0.46 | 5000 |
| Fast-SNN | YOLOv2(ResNet34) | 76.16 | 76.05 | -0.11 | 15 | YOLOv2(ResNet34) | 46.96 | 46.40 | -0.56 | 15 |
| SpikeZIP-N | YOLOv3 | **77.55** | **77.48** | **-0.07** | 15 | YOLOv3 | **52.10** | **52.20** | **+0.10** | 15 |

replace the LeakyReLU Xu et al. (2020) in the backbone of YOLOv3 with ReLU function and utilize the conversion pipline of SpikeZIP shown in fig. 1 to obtain the SNN for object detection. The comparison between the converted SNN and other conversion-based works is shown in table 6. Our SpikeZIP achieves higher mAP than the Fast-SNN Hu et al. (2023) with the same time-step.

### 4.5 ENERGY CONSUMPTION ANALYSIS

**Neuron level.** As the foundation unit of SpikeZIP, ST-BIF neuron differs from the classic IF neuron in its bipolar output spike and extra spike tracer. We perform power analysis of neuron variants with CMOS 65nm technology node, where the hardware evaluation is given in table 7. Both *mem.* and spike tracer are multi-bit registers to buffer the membrane potential and neuron firing record respectively. Thanks to the latency compression in SpikeZIP which compresses the SNN inference time-steps within 11, the 4-bit spike tracer register is sufficient for ST-BIF neuron which consumes 1.22% more power than the IF counterpart.

**Model level.** We compare the energy consumption of the SNN version of ResNet20 converted from QCFS Bu et al. (2023) and SpikeZIP-PR using CIFAR100. We take the synaptic operation (SOP) introduced in Merolla et al. (2014) to represent the basic operation numbers to infer one image in SNN, while the ANN counterpart uses floating-point operations (FLOP). For the energy consumption per operation, we use 77fJ/SOP and 12.5pJ/FLOP reported from the ROLLS neuromorphic processor Qiao et al. (2015). We utilize the energy consumption estimation method used in QCFS Bu et al. (2023), where the results are reported in fig. 7. We can observe that the energy consumption of SNN is much lower than the ANN (1.02mJ per inference). Compared to QCFS, SpikeZIP also consumes less energy where the energy saving is enlarged when $T > 8$, due to the SNN generated by SpikeZIP enters the equilibrium state.



| neuron model | *mem.* (#bit) | tracer (#bit) | power ($\mu$W) |
|---|---|---|---|
| IF | 32 | *n/a* | 324.00 |
| ST-BIF | 32 | 4 | 327.97 |
| ST-BIF | 32 | 8 | 361.50 |
| ST-BIF | 32 | 16 | 428.62 |
| ST-BIF | 32 | 32 | 565.29 |

Table 7: The power consumption of IF neuron and ST-BIF neuron with spike tracers of different bit-width.

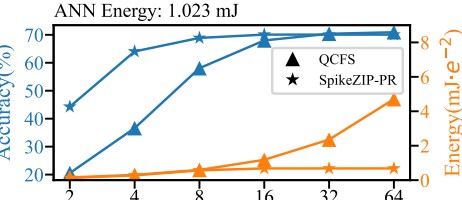

Figure 7: accuracy and energy v.s. time-step $T$ for SNN using QCFS Bu et al. (2023) and SpikeZIP-PR respectively.

## 5 CONCLUSION AND OUTLOOK

SpikeZIP constructs a framework for obtaining a high performance SNN, supported by comprehensive experiments and rigorous proof in this work. We anticipate the SpikeZIP as a foot-stone for future investigations of brain-inspired SNN, which bridges and inherits the existing research of deep learning to the paradigm of neuromorphic computing. For compact models where PET and RCR are applicable, SpikeZIP empowers its converted SNN to achieve the state-of-the-art performance. For large models where retraining or fine-tuning are not feasible, the mathematical equivalence established in SpikeZIP can also provide a promising approach of direct conversion, in combination with the post-training quantization technique for Q-ReLU.

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
