# OpenReview forum: "SpikeZIP: Compressing Spiking Neural Network with Paths-Ensemble Training for Optimized Pareto-front Performance"
_ICLR.cc/2025/Conference — Submitted to ICLR 2025_

### Official Review · Reviewer_DNQJ · 2024-10-22

**Soundness:** 2
**Presentation:** 3
**Contribution:** 2
**Rating:** 3
**Confidence:** 4

**Summary:**

This paper employs a novel training framework, which first trains a customized Quantized Artificial Neural Network (QANN) and then converts it into a Spiking Neural Network (SNN). During the QANN training process, the authors propose a PET that effectively improves model performance while reducing the required inference time steps, demonstrating a degree of innovation. Furthermore, the authors' method achieves theoretical equivalence between the QANN and the converted SNN, which is a valuable contribution to SNN conversion methodologies.

**Strengths:**

## originality
This paper demonstrates a high level of originality, particularly in the design of the PET module, the reconsideration of SEW-ResNet in the ANN2SNN method, and the proof of equivalence between QANN and the converted SNN.
## quality
The paper is exceptionally well-written, with clear organization and concise, comprehensible content. The description of the proposed method is both lucid and accurate.
## clarity
The overall structure of the paper is very clear. The Introduction and Related Works sections, in particular, provide an easily understandable overview of the current state of the ANN2SNN field. Additionally, the decision to present Table 2 before its explanation is an effective design choice that enhances reader comprehension.
## significance
This paper has scientific significance as it reinforces certain standards in ANN2SNN conversion, such as the equivalence between ANN and the converted SNN. It also provides a novel solution for training low-latency, high-accuracy SNNs.

**Weaknesses:**

1. Some descriptions are unclear or confusing:
- Lines 17-20: "Nevertheless, existing works fail at providing theoretic equivalence between Quantized-ANN (QANN) and its converted SNN, while the SNN accuracy at small time-step (i.e., Pareto-frontier) can be further improved."
The last row in Table 1: "Quantization levels in major-path and i-th sub-path."
- Lines 119-121: There are punctuation errors. It might be better to revise as: "ana. denotes analog input encoding; mem calib. is short for membrane calibration; and eq. stands for equivalence."
- Lines 250-252: The description is confusing and it's difficult to understand without reading subsequent content that convolutional layers in major-path and sub-paths share weights.
- Lines 327 and 328 mention the use of 'triangle symbol' and '-' symbol in the table to indicate SNN reaching equilibrium and most spikes not reaching SNN output, respectively. However, these symbols are not visible in Table 3. Is this an oversight?
2. In Table 4, the originally reported result for MS-ResNet34 shows timesteps equal to 6. The authors may have made an error in data collection here.
3. ﻿Lines 405-407 contain a descriptive error. The inference time steps of SNN should not be less than that of ANN with T=1.
4. The method of first training QANN and then converting it to SNN has already been proposed [1], so I don't consider this an innovation.
5. The motivation for proposing PET is insufficient and seems arbitrary. It's unclear what led the authors to choose this multi-scale parallel method. I understand that PET doesn't introduce additional inference computations for the conversion method, but the authors should provide ablation experiments to demonstrate the superiority of using PET compared to networks with only one stream. I believe that merely presenting data in Figure 6 without any analysis is insufficient to justify the necessity of PET.
6. Even though the authors state that their purpose for using the SEW residual structure differs from that in directly trained SEW-ResNet, I don't think this can be considered an innovation.
7. The effectiveness of this work is not satisfactory. In Table 3, it's difficult to see the superiority of the proposed method in terms of conversion or quantization. Regarding classification accuracy and reduction of inference time steps, this work's performance is inferior to some recent representative works not mentioned in this paper, such as [2].


[1]You K, Xu Z, Nie C, et al. SpikeZIP-TF: Conversion is All You Need for Transformer-based SNN[J]. arXiv preprint arXiv:2406.03470, 2024.

[2] Lv L, Fang W, Yuan L, et al. Optimal ANN-SNN Conversion with Group Neurons[C]//ICASSP 2024-2024 IEEE International Conference on Acoustics, Speech and Signal Processing (ICASSP). IEEE, 2024: 6475-6479.

**Questions:**

1. The paper repeatedly mentions the use of 'analog input encoding', but I am unfamiliar with this encoding method. Upon reviewing the authors' cited references, I found that 'analog input encoding' is actually 'direct coding'. Given this, why not use the more widely recognized term 'direct coding' in the paper? Have previous papers referred to direct coding in this manner? Alternatively, is my understanding incorrect, and is there a distinction between 'analog input encoding' and 'direct coding'?
﻿
2. How is the energy consumption calculated in the right half of Figure 7? To my knowledge, the SEW residual structure is not spike-driven, which means that floating-point multiplications exist in the first convolutional layer after the residual connection. This would lead to higher overall energy consumption and computational load for the network. Did the authors consider this when calculating energy consumption? Specifically, did they treat the operations in the first convolutional layer after each residual connection as floating-point operations in their calculations?

---

> ### Comment · Reviewer_DNQJ · 2024-11-26
> **We sincerely appreciate the reminder from the ICLR committee. However, it appears that the authors are not inclined to engage in discussions regarding this particular paper.**
>
> We sincerely appreciate the reminder from the ICLR committee. However, it appears that the authors are not inclined to engage in discussions regarding this particular paper.

---

### Official Review · Reviewer_o14M · 2024-10-23

**Soundness:** 1
**Presentation:** 2
**Contribution:** 2
**Rating:** 3
**Confidence:** 5

**Summary:**

This paper focuses on methods for converting ANN to SNN without loss of accuracy. The authors believe that existing work faces two problems: first, the failure to provide a theoretical equivalence between quantized ANN (QANN) and the converted SNN; and second, there is still significant room for performance improvement when using shorter time steps. To address these issues, this paper proposes a new conversion framework called SpikeZIP, which utilizes the two-step conversion process from ANN to QANN, and then SNN. More specifically, the SpikeZIP framework incorporates several components, including SNN Friendly Morphing (also, RCR), Paths-Ensemble Training (PET), and others. The experimental results demonstrate that SpikeZIP achieves superior performance.

I acknowledge the substantial effort involved in this paper; however, after multiple readings, I still have many unresolved questions and hope the authors can provide further clarification.

**Strengths:**

1. The abstract and introduction of the article are logically clear and well-structured.
2. The paper effectively mitigates the gap in theoretical equivalence between QANNs and SNNs.
3. The experimental results clearly demonstrate the effectiveness of the proposed SpikeZIP framework, highlighting substantial performance improvements.

**Weaknesses:**

This paper is not innovative enough, the RCR design has problems, and the PET is not convincing. More specifically:
1. There is a published paper [1] utilizing the same approach, i.e., ANN-QANN-SNN two-step conversion, for the objective of lossless conversion. This published paper has already been applied to complex Transformer architectures, while SpikeZIP has only been validated on simpler convolutional structures. It would be beneficial for authors to provide more specific details regarding how the approach in [1] compares with SpikeZIP. For instance, what are the key similarities and differences between the two? Additionally, are there any aspects of SpikeZIP that are novel compared to [1]?
2. The residual connection re-routing (RCR) method is originally proposed to be SNN-friendly; however, it clearly undermines this objective. As demonstrated in the SNN basic block in Figure 3, the vanilla structure preserves the spike-driven nature of SNNs. However, the output of the proposed SZIP is an integer, which disrupts the important spike-driven advantage of SNNs. So, why do authors think the designed RCR maintains SNN-friendliness? Additionally, the proposed RCR has already been extensively explored in SEW-ResNet (such as ADD, AND, and IAND).
3. The purpose of PET design is to improve performance. However, the author has not provided a clear explanation as to why dividing the backpropagation into three paths is necessary for achieving this performance improvement. The authors are encouraged to provide a more detailed justification for the three-path design of PET.

Ref: [1]: SpikeZIP-TF: Conversion is All You Need for Transformer-based SNN.

**Questions:**

1. One of the contributions of SpikeZIP is the theoretical equivalence between QANN and SNN. However, as shown in Table 2, this equivalence has already been explored in EMRS and Radix. Therefore, the authors are encouraged to add discussion to clarify how their theoretical equivalence differs from or improves upon the existing work in EMRS and Radix.
2. I am unclear about the encoding method presented in Section 3.2, particularly regarding the meaning of $\delta v$. Could a small figure be provided to illustrate how this encoding method differs from the previously used direct encoding? Additionally, what are the advantages of this new encoding method?
3. In the proposed PET, what is the basis for dividing the three pathways? Why do Conv/Linear, Classifier, and Ensemble loss share parameters? What are the benefits of this parameter sharing?
4. The footnote of RCR in Sec 3.3 states that “SEW-ResNet adopts RELU before addition”. As far as I know, it is not used.
5. Does SNN Friendly Morphing refer to an architecture that is more conducive to lossless conversion? If so, how does the author solve the non-spike-driven problem introduced by this architecture? This is a question that deserves attention.
6. The writing of the paper is quite rough, and it is recommended that the authors revise it carefully. For example:

(1) The keywords are included in the abstract.

(2) The abbreviation "ST-BIF" appears for the first time in the contribution section but is not explained.

(3) In Tab.1, the capitalization of the first letters in the symbol descriptions is inconsistent.

(4) Inconsistent usage, such as "ReLU" and "RELU."

(5) "Pareto frontier" is a technical term, but its meaning is not explained.

(6) In Tab. 1, the explanation for $n, n_{mp}, n_{sp}, i$ is given as "Quantization levels in major-path and the i-th sub-path", which is unclear. It is recommended that the authors explain each symbol separately.

(7) Fig. 3 appears earlier in the paper than Fig. 2.

(8) The caption in Fig. 3 uses the past tense.

(9) Does "SZIP" in Fig. 3 represent the proposed SpikeZip? Its appearance lacks clarity.

---

### Official Review · Reviewer_8YNX · 2024-10-28

**Soundness:** 2
**Presentation:** 2
**Contribution:** 2
**Rating:** 3
**Confidence:** 5

**Summary:**

The paper presents SpikeZIP, a novel framework aimed at optimizing Spiking Neural Networks (SNNs) through a two-step conversion process from Artificial Neural Networks (ANNs). By employing paths-ensemble training (PET) and achieving model-level equivalence between Quantized ANNs (QANNs) and SNNs, SpikeZIP significantly improves the trade-off between accuracy and latency (measured in time steps). SpikeZIP demonstrates competitive performance on CIFAR100 and ImageNet, where it outperforms state-of-the-art (SOTA) methods in both accuracy and energy efficiency.

**Strengths:**

1. The proposed paths-ensemble training technique and conversion framework establish a meaningful connection between QANNs and SNNs, improving the model’s accuracy-latency trade-off. Theoretical proofs and practical implementations of equivalence are valuable contributions to ANN-SNN conversion research.
2. SpikeZIP focuses on reducing the computational cost and energy consumption, which is essential for neuromorphic hardware applications. The paper’s comparative analysis, including power consumption metrics, highlights the energy benefits of the proposed framework.
3. The demonstrated adaptability of SpikeZIP to different network architectures (e.g., ResNet, VGG, YOLOv3) enhances its utility and applicability in real-world edge-computing tasks.

**Weaknesses:**

1. While paths-ensemble training is central to the SpikeZIP framework, a more in-depth explanation with specific pseudocode would make the method more accessible to readers. The authors should include a pseudocode algorithm or flowchart for the PET method in the paper or appendix. Additionally, a direct comparison between PET and conventional quantization-aware training is required, highlighting the key differences and advantages.
2. The paper would benefit from explicitly listing the assumptions and constraints under which the QANN-SNN equivalence holds. This would clarify the conditions necessary for achieving equivalence and enable more targeted reproduction. The authors should provide a clear list or table of the necessary conditions and constraints for QANN-SNN equivalence. This would be very helpful for readers trying to implement or build upon this work.
3. Although the focus is on conversion-based methods, a more extensive comparison with learning-based SNN training approaches could better contextualize the advantages of SpikeZIP, particularly in terms of energy efficiency and latency. It will be beneficial to include a comparative table or figure showing SpikeZIP's performance against leading learning-based SNN methods on standard benchmarks, focusing on both accuracy and energy efficiency metrics.

**Questions:**

1. Could you provide a more detailed explanation or pseudocode for the paths-ensemble training method? Specifically, how does PET differ from traditional quantization-aware training, and what are the specific steps involved?
2. What specific conditions are necessary for the QANN-SNN equivalence to hold? Are there particular constraints on the model architecture or quantization levels that should be met to ensure theoretical equivalence?
3. Have you tested SpikeZIP on any architectures beyond VGG, ResNet, and YOLOv3? Extending to other architectures, especially recurrent or transformer-based SNNs, could showcase SpikeZIP’s generalizability.
4. How does SpikeZIP’s performance in terms of accuracy and energy efficiency compare with SNNs trained directly through learning-based methods? This comparison could highlight the relative benefits of conversion-based approaches like SpikeZIP.
5. Have you considered implementing and testing SpikeZIP on actual neuromorphic hardware, such as Intel Loihi? This would help validate the energy efficiency and latency improvements under real-world conditions.
6. Could you elaborate on the impact of different quantization levels on SpikeZIP’s performance? A breakdown of how accuracy and latency trade-offs vary across quantization levels would provide more insights into optimizing QANN configurations.

---

### Official Review · Reviewer_dRSY · 2024-11-04

**Soundness:** 3
**Presentation:** 3
**Contribution:** 2
**Rating:** 3
**Confidence:** 5

**Summary:**

In this work, the authors proposed SpikeZip, which is an ANN-to-SNN conversion framework for low-latency inference of SNNs. SpikeZip consists of SNN-friendly morphing, paths-ensemble training, operator fusion, and neuron replacing. Furthermore, they provided theoretical equivalence between QANN and SNN. To validate the effectiveness of the proposed methods, the authors conducted experiments on various models (VGG and ResNet) and datasets (CIFAR-100 and ImageNet).

**Strengths:**

- Proposing a method of ensemble training with various quantization levels to reduce conversion error.
- superior Pareto-front (higher accuracy and lower time step)
- Performance comparison with recent works.

**Weaknesses:**

Despite the improved performance of this work, many concerns must be resolved for publication.

- Lack of novelty
    - Most of the components of SpikeZIP presented in this paper have been widely applied in other related studies.
    - What are the distinct features of SNN-friendly morphing compared to other studies?
    - Operator fusion and neuron replacing have also been widely applied in ANN-to-SNN conversion. What are the differences?
    - Input and bias encoding have also been widely used. What are the differences?
    - Residual connection re-routing is the same as SEW-ResNet as mentioned in the manuscript. The novelty of the proposed method does not exist just because of the different purpose. It is appropriate to judge that the existing method was applied. Additional explanations of the differences based on experimental results or theoretical analyses are required.
    - The theoretical proof in section 3.5 does not seem to be much different from the content published in the existing study. What is the difference from the proof in [2]?
- Lack of specific analysis of the proposed method
    - Why does the inference accuracy of the converted SNN increase when PET is applied? A specific and clear analysis is needed. Also, why were three paths {2, 4, 8} used? What effect does the number and composition of paths have on the conversion?
- Lack of logical connection between the theoretical proof and the proposed method
    - What relationship does the theoretical proof have with the proposed method? The proposed methods can be used without theoretical proof, and the method does not use the results of the theoretical proof.
    - Generally, T_{eq} > T_{off}, but how does T_{eq} affect the experimental results in the experiment?

- Manuscript revision
    - Last part of abstract
    - Missing definition of abbreviations
        - QANN, SOTA
        - line 183: LSQ-based
    - typos
        - line 39 - “))” → ‘)’
        - line 87
    - Duplicate notation
        - V^{in} in equations 1 and 7 have different meanings but are used in the same notation.
    - missing contents
        - line 327: The triangle symbol is not in the table. What does it mean?
    - Readability
        - The font size of Table 3 is too small to read.
        - The notation is concentrated in Table 1, so it is a bit inconvenient to look at the equation.
        - It would be better to switch the order of Figures 2 and 3. Section 3.3 is connected to Figure 3, and Section 3.4 is connected to Figure 2.
    - line 85: ST-BIF neuron reference required
    - line 347: no appendix provided

**Questions:**

- Title - why SpikeZip?
- about neuron type (ST-BIF)
    - According to Equation 1, for ST-BIF operation, the spike and spike count of the previous time step affect the neuronal dynamics. Is this neuron type applicable to neuromorphic hardware? What is the overhead compared to IF, LIF?
    - Why use ST-BIF instead of the commonly used IF, LIF? What happens to the conversion results when IF, LIF are used?
    - It seems that a more detailed explanation of the role of spike tracer (St) is needed.
- about PET
    - Could you compare the PET with the below paper?
        - Stochastic Precision Ensemble: Self-Knowledge Distillation for Quantized Deep Neural Networks (AAAI-21)
- about ablation studies
    - There is a lack of analysis of the methods proposed in ablation studies. Most of them only suggest experimental results obtained by parameter sweep.
    - It seems that the PET quantization level is heuristically determined, so there seems to be room for further improvement.
- Table 3
    - Compared to other conversion papers, QANN and ANN are mixed and shown in “(Q)ANN” column. When looking at the accuracy, there are cases where SNN has higher accuracy than ANN, but it is not clear whether this is only the case for QANN or whether this is also the case among ANNs, so it would be better to show QANN and ANN separately.
    - In (b), when T=16, spikeZIP-PR has lower accuracy than offset. Why is that?
- Table 4
    - Compared to direct training methods, there is no explanation of how this paper's method (SpikeZIP-PR) was trained.
    - What about SpikeZIP-P instead of SpikeZIP-PR?
    - What about ResNet and ImageNet?
- Fig. 5
    - How about SpikeZIP-PR in the case of L1-Norm comparison?
    - Why is SpikeZIP-PR compared in Table 4, and SpikeZIP-P compared in Fig. 5? - This confusing comparison setting raises doubts about the experimental results of this study.
- line 272: why? Can you show us specific experimental results?
- lines 284, 285: This does not seem appropriate in the given context.
- lines 417, 418: Why does the intrinsic error of GPU exist? Why is the error of the L1-norm caused by this 0.5? It is difficult to easily agree with the explanation of the intrinsic error of GPU. Please explain it in more detail.

---

> ### Comment · Reviewer_dRSY · 2024-11-28
>
> If the authors require further discussion, please post a response.

---

### Official Review · Reviewer_eFF1 · 2024-11-05

**Soundness:** 3
**Presentation:** 3
**Contribution:** 3
**Rating:** 6
**Confidence:** 2

**Summary:**

This paper presents SpikeZIP, a novel framework for converting artificial neural networks (ANNs) to spiking neural networks (SNNs) with improved performance.

**Strengths:**

The paper provides a rigorous proof of theoretical equivalence between Quantized ANNs (QANNs) and their converted SNNs at the model level. This addresses a gap in existing literature and provides a solid foundation for the conversion process.

The proposed paths-ensemble training (PET) algorithm considers SNN temporal information when fine-tuning QANNs.

Impressive results on ImageNet, outperforming existing methods in terms of accuracy

**Weaknesses:**

While the paper claims superior performance, it would be beneficial to see a more comprehensive comparison with other recent SNN conversion methods. For e.g. [1] has imagenet accuracy with <6 timesteps. Can the authors comment why is their latency so high as compared to [1]? Similar to [1], there are many other works that look into efficient and fast inference with low latency in SNNs even with transformers [2-4]. Can the authors comment on it? What about video data [5]?

The individual impact of each component in the SpikeZIP framework should be provided.

Will the technique scale to transformer architectures [2-4]?

[1] Li, Yuhang, et al. "Seenn: Towards temporal spiking early exit neural networks." Advances in Neural Information Processing Systems 36 (2024).

[2] Yao, Man, et al. "Spike-driven transformer." Advances in neural information processing systems 36 (2024).

[3] Zhou, Chenlin, et al. "Spikingformer: Spike-driven residual learning for transformer-based spiking neural network." arXiv preprint arXiv:2304.11954 (2023).

[4] Lee, Donghyun, et al. "Spiking Transformer with Spatial-Temporal Attention." arXiv preprint arXiv:2409.19764 (2024).

[5] Xiao, Shiting, et al. "ReSpike: Residual Frames-based Hybrid Spiking Neural Networks for Efficient Action Recognition." arXiv preprint arXiv:2409.01564 (2024).

**Questions:**

See weaknesses above

---

### Meta-Review · Area_Chair_HuEg · 2024-12-22

**Metareview:**

This paper introduces a framework called SpikeZIP for efficient conversion of artificial neural networks (ANN) into spiking neural networks (SNN). The author's motivation is to solve the problems of lack of theoretical equivalence between quantized ANN (QANN) and SNN and insufficient performance of short time steps in existing methods. The authors optimized the QANN to SNN conversion process by introducing a two-step conversion algorithm of paths-ensemble training (PET) and other techniques. Experimental results show that SpikeZIP achieves higher accuracy and lower latency on ImageNet and CIFAR-100, outperforming current state-of-the-art methods.

After the initial reviewing stage, 5 reviewers rate 3, 3, 3, 3, and 6, respectively. Most reviewers agree that SpikeZIP presents a theoretically rigorous proof of equivalence between QANN and SNN and improves the accuracy-latency trade-off of the model, which is the main advantage of this approach. However, there are some criticisms. Reviewers eFF1 and dRSY believe that the performance improvement of SpikeZIP is not significant enough compared with existing methods, some components lack novelty, and there is a lack of comprehensive comparison with other related research. Reviewers 8YNX and DNQJ noted that the description and discussion of PET is not transparent enough. In addition, the reviewers also summarized many detailed issues, including experimental design, result presentation, and content presentation. Overall, the reviewers suggested strengthening the comparative analysis with existing methods, adding more detailed explanations, and optimizing the presentation of the manuscript. I think these suggestions are reasonable and the author should improve the article accordingly.

Since there was no improvement in ratings after the rebuttal period, I think this paper needs further revision. Therefore, the final decision was made to reject the paper.

**Additional Comments On Reviewer Discussion:**

The author did not discuss the reviewers' suggestions during the discussion stage.

---

### Decision · Program_Chairs · 2025-01-22

Reject